# Formulation Studies with Cyclodextrins for Novel Selenium NSAID Derivatives

**DOI:** 10.3390/ijms25031532

**Published:** 2024-01-26

**Authors:** Sandra Ramos-Inza, Cristina Morán-Serradilla, Leire Gaviria-Soteras, Arun K. Sharma, Daniel Plano, Carmen Sanmartín, María Font

**Affiliations:** 1Department of Pharmaceutical Sciences, University of Navarra, Irunlarrea 1, E-31008 Pamplona, Spain; sramos.2@alumni.unav.es (S.R.-I.); cmoran.3@alumni.unav.es (C.M.-S.); lgaviria@alumni.unav.es (L.G.-S.); dplano@unav.es (D.P.); mfont@unav.es (M.F.); 2Instituto de Investigación Sanitaria de Navarra (IdiSNA), Irunlarrea 3, E-31008 Pamplona, Spain; 3Department of Pharmacology, Penn State Cancer Institute, CH72, 500 University Drive, Hershey, PA 17033, USA; asharma1@pennstatehealth.psu.edu

**Keywords:** cyclodextrin, selenium, NSAID, solubility, formulation, molecular modeling

## Abstract

Commercial cyclodextrins (CDs) are commonly used to form inclusion complexes (ICs) with different molecules in order to enhance their water solubility, stability, and bioavailability. Nowadays, there is strong, convincing evidence of the anticancer effect of selenium (Se)-containing compounds. However, pharmaceutical limitations, such as an unpleasant taste or poor aqueous solubility, impede their further evaluation and clinical use. In this work, we study the enhancement of solubility with CD complexes for a set of different nonsteroidal anti-inflammatory drug (NSAID) derivatives with Se as selenoester or diacyl diselenide chemical forms, with demonstrated antitumoral activity. The CD complexes were analyzed via nuclear magnetic resonance (NMR) spectroscopic techniques. In order to obtain additional data that could help explain the experimental results obtained, 3D models of the theoretical CD-compound complexes were constructed using molecular modeling techniques. Among all the compounds, **I.3e** and **II.5** showed a remarkable increase in their water solubility, which could be ascribed to the formation of the most stable interactions with the CDs used, in agreement with the in silico studies performed. Thus, the preliminary results obtained in this work led us to confirm the selection of β and γ-CD as the most suitable for overcoming the pharmaceutical drawbacks of these Se derivatives.

## 1. Introduction

It is a well-known fact that a wide number of state-of-the-art treatments present several drawbacks, such as a year-on-year increase in drug resistance, detrimental side effects, decreased quality of life of the patients, and reduced treatment adherence. Consequently, developing more effective and safer drugs remains a challenging and critical goal for the scientific community. In this regard, a good deal of effort has been devoted to the use of cyclodextrins (CDs). These cyclic oligosaccharides are formed by the bacterial metabolization of starch and are composed of a number of glucopyranose units linked by α-1,4-glucosidic bonds [1]. They present a truncated cone structure with a hydrophilic outer surface and a relatively hydrophobic central cavity. The primary hydroxyl (OH-) groups are located on the upper rim of this structure, whereas the secondary ones are on the lower rim. The most commonly available types of CDs are α-, β-, and γ-CDs, comprising six, seven, and eight D-glucose units, respectively (Figure 1) [2].

A cohort of studies supports the indisputable promise of CDs due to their unique combination properties [3,4,5,6]. They can serve as active compounds’ carriers while enhancing some of their features (e.g., solubility, hydrolytic stability, and bioavailability) [2,7]. It has also been reported that the physicochemical properties of these molecules depend on the structure of the attached substituents along with their location and number on the CD’s framework [8].

The role of the element selenium (Se) in preventing cancer incidence is well known [9]. Alongside traditional therapies as an adjuvant, Se supplementation was demonstrated to enhance the efficacy of standard chemotherapeutic drugs and limit the side effects without reducing the effectiveness of the therapy [9,10,11]. Accordingly, Se-containing compounds have attracted attention in the field of drug discovery for cancer therapy [12,13,14] by acting through different mechanisms of action [12,15,16].

Although traditionally known for their analgesic, anti-inflammatory, and antipyretic prescriptions [17], in recent years, nonsteroidal anti-inflammatory drugs (NSAIDs) have been recognized for their efficacy in preventing and treating cancer at different stages of the progression of this pathology [18,19,20,21]. In this context, the optimization of NSAID scaffolds by means of several different chemical modifications in order to develop more effective agents with new potential mechanisms of action has been widely explored [22,23]. Among the different strategies proposed, the incorporation of Se has been reported as an appealing approach for obtaining new NSAID-based chemotherapeutics [24,25,26].

Likewise, several studies reported the impact of CD incorporation on the biological effects of NSAIDs and their derivatives and supported the use of CDs as appealing drug delivery systems capable of improving the pharmacological and biopharmaceutical properties of this type of drug [27,28]. Moreover, the use of CDs not only emerges as an interesting option for formulating NSAIDs but also for overcoming the usual drawbacks of Se-containing compounds, such as poor solubility and toxicity [29]. In this regard, the incorporation of CDs to form inclusion complexes (ICs) has been demonstrated to be a valid approximation to improve the therapeutic efficacy of selenocompounds due to an enhancement of their pharmacological properties [30].

Thus, considering the chemotherapeutic activity of NSAIDs and the anticancer profile of Se and Se-bearing derivatives, we developed two series of Se-NSAID analogs depending on the chemical form in which the chalcogen was incorporated into the different scaffolds. The two series were designed based on the inclusion of Se either as selenoester (series I) or diacyl diselenide (series II). The obtained compounds were further reported as potent and selective agents for the treatment of breast [31] and colon [32] cancers, respectively.

In Figure 2, all the Se-NSAID derivatives included in this work are gathered. Given the large number of compounds that comprised series I, the five most bioactive compounds (**I.3e** and **I.4a–e**) were selected alongside the entirety of series II (**II.1–5**) for further investigations.

Hence, in line with our aim to develop new antitumoral drugs, the objective of this work is the formulation of these series of bioactive Se-NSAID derivatives with β-CDs and γ-CDs. Thus, the possible exhibition of an increase in the water solubility of the complexes in comparison with the isolated compounds was evaluated as a first approximation to test the ability of these derivatives to be encapsulated inside this type of carrier to improve their bioavailability and biological properties. Concurrently, a preliminary in silico analysis was carried out by means of molecular modeling with the Se-NSAID derivatives and CDs in order to explain the experimental results observed. By employing a docking strategy where the corresponding CD was treated as the receptor and the Se-NSAID derivatives as the ligands, the potential formation of such complexes was predicted.

## 2. Results and Discussion

### 2.1. Formulation of Se-NSAID Derivatives with β-CDs and γ-CDs

For all the Se-NSAID derivatives, the formation of inclusion complexes either with β- or γ-CDs was evaluated by quantitative proton nuclear magnetic resonance (^1^H-NMR) spectroscopy (Appendix A). The election for such CDs is based on the fact that β-CD is one of the most frequently used CDs for encapsulating NSAIDs [28]. Additionally, given the size of the studied molecules, a larger CD such as γ-CDs was also evaluated. The solubilities calculated for the compounds alone and after the addition of the corresponding CD in an aqueous medium are displayed in Table 1.

According to the data depicted in Table 1, it can be observed that all the studied Se-NSAID derivatives exhibited poor water solubility, whereas their corresponding inclusion complexes with β- or γ-CDs resulted in an enhancement of this property. Among them, derivatives **I.3e** and **II.5** showed a remarkable improvement in their solubility after the exposure to both CDs. Of note, **II.5** is the most soluble Se-NSAID derivative overall, with an increase of up to 250- (5.42 × 10^−3^ M with β-CD) and 350-fold (7.50 × 10^−3^ M with γ-CD) in the concentration values obtained, respectively. Interestingly, this Se-NSAID derivative **II.5** had already stood out among the diacyl diselenide series for its potent anticancer properties demonstrated both in vitro and in vivo [32], so its incorporation into CDs could be a feasible starting point for further drug development.

Given the results displayed in Table 1, we decided to undergo a molecular modeling study to further explore the formulation with CDs. According to our initial working hypothesis, by means of a docking approach, in which the different CDs were selected as receptors (host) and the target molecules were considered as the theoretical ligands, the possible formation of complexes could be predicted and, subsequently, their stability could be evaluated by means of a molecular dynamic strategy. We propose that the greater the availability of formation of a given complex and its stability, the greater the possibility of modulating the aqueous solubility of the compound under study.

Regarding the construction of the 3D models of the studied Se-NSAID derivatives and their CDs complexes, several templates from the Cambridge Structural Database (CSD) [33] were used. The selected models were imported into the MOE2022.02 workspace [34], curated (charges and protonation state, pH 7), and minimized. In order to have a further insight into the characteristics of the binding site of the CDs, their potential electrostatic and interaction maps were obtained.

The study of their conformational trajectories and the detection of the lowest energy conformation for the compounds were analyzed due to the high conformational freedom ascribed to the side chains of the structures and their possible influence on the inclusion behavior of the above-mentioned Se-NSAID compounds. Therefore, a preliminary analysis of the conformational behavior both in implicit and explicit solvent (water) conditions was carried out through a LowModeMD approach [35] after the theoretical initial 3D models were constructed.

The docking study was carried out by applying the Dock algorithm implemented in the MOE2022.02 suite (AMBER10:EHT force field), with explicit solvent (water) conditions which were simulated by applying the solvate routine implemented in the MOE2022.02 suite (spherical droplet mode).

### 2.2. 3D Models and Analysis of the β- and γ-CDs

For the construction of the 3D models of the selected CDs, the reference models AJUVEG [36] and CIWMIE [37] were downloaded from CSD as templates of the β- and γ-CDs, respectively, and incorporated into the MOE2022.02-suite workspace with water as implicit solvent. The models were curated (charges and protonation state, pH 7) and minimized. Additionally, the potential electrostatic and interaction maps were obtained in order to define complementary information of the binding site (e.g., donor, acceptor, hydrophobic area features, etc.) that could be useful for understating the host–guest interactions with the evaluated Se-NSAID compounds. The obtained models for each CD can be found in Figure 3.

For both models (Figure 3a,b), once the initial conformation was constructed, a minimization protocol was applied, and the lowest energy conformation was studied. Thereby, the corresponding maps of the inner cavity were obtained and are displayed in Figure 4.

Concerning the data obtained for the β-CD model (Figure 4a), an area that presented a considerable size that was distributed along the edges of the minor face was shown. This area formed a wall of hydrogen bridge donors which was constructed based on the hydroxyl groups located at position 6 of the rings. Only small surfaces with acceptor features were distributed along the external surface. In the inner cavity, a discontinuous area where the inwardly oriented hydroxyl moieties acted as donors was detected, whereas a continuous hydrophobic region was located in the center of this cavity.

Likewise, the γ-CD data (Figure 4b) were proven to have a similar behavior as the other CD, albeit a wider acceptor zone was detected in the outer surface of the structure. Moreover, the inner hydrophobic area was larger but less continuous than for the β-CD model.

### 2.3. 3D Models and Analysis of the Se-NSAID Derivatives

In the case of the organoseleno compounds, the CSD references for the NSAID parent scaffolds (Figure 5) were imported on the MOE2022.02 workspace and modified with the Builder tool in order to obtain the exact structures of the studied analogs. After that, they were minimized with the implemented MOPAC engine (PM3 semi-empirical approach [38]).

The structural analysis of these compounds allowed for the identification of four different scaffolds that could be used to classify them: naphthalene (**I.3e** and **II.3**), indole (**I.4a–e** and **II.2**), benzene (**II.1** and **II.5**), and diphenylketone (**II.4**). These are π-aromatic systems that imply the existence of flat, rigid zones which confer a nonpolar character to the molecules.

As complementary data, and in order to select the theoretically most adequate CD, the values of logP and van der Waals volume and surface were calculated and are displayed in Table 2.

The above-mentioned data regarding logP (Table 2) confirmed their hydrophobic character, especially for the derivatives of series II. In view of the former, and considering the volume and surface values obtained, it can be concluded that the use of β- and γ-CDs as carriers of these Se-NSAID analogs could be considered as a good strategy by which to increase their solubility in water.

On the other hand, in Figure 6 and Figure 7, the conformational behavior and the lowest energy conformations for both series of Se-NSAID derivatives are outlined.

Considering the conformational behavior, the lateral chains and the presence of the diphenylketone core (an oxo group surrounded by two phenyl rings, joined by single bonds) of some of these Se-NSAID analogs could interfere with possible effective interactions between CDs and the guest molecules due to their broad conformational trajectory.

In fact, this conformational freedom of the analyzed derivatives is exemplified in the derivatives of series II, in which the diacyl diselenide group present in compounds **II.1–5** could increase the possibility of adopting different conformations. Furthermore, another fact that should be pointed out is that the solvation conditions used (performed with implicit or explicit solvent) have an influence on the conformational behavior of the derivatives (Figure 6 and Figure 7). In this regard, the results showed that the greatest difference between the lowest energy conformations obtained for **I.3e** and **I.4a–e** in both tested solvation conditions (Figure 6) was related to the arrangement of their side chains. In the implicit solvent conditions, the side chain folded slightly towards the scaffold, whereas under explicit solvent conditions, it unfolded away from it.

Regarding the derivatives of series II (Figure 7), the data obtained under implicit solvent conditions showed how the presence of the central diselenide moiety (Se-Se) was apparently responsible of the C-fold conformation preferred by these compounds, in which the side rings are stacked and can form intramolecular π–π interactions. Surprisingly, this conformation was not detected for derivative **II.5**. However, under explicit solvent conditions, it was observed that the preferred conformation changed to extended ones, in which the intramolecular interactions disappeared. Thus, the elements that theoretically could interact with the active site of the CDs acquired a different spatial distribution.

### 2.4. Docking Data

With the aim of corroborating the experimental results obtained in the solubility assay and shedding light on the possible underlying mechanism of complexation between the Se-NSAID derivatives and CDs, we carried out a docking study. In view of the aforesaid relationship between the conformational behavior of the compounds and the solvent conditions used, we decided to adopt an explicit solvent strategy in which the water conditions were simulated by applying the solvate routine (spherical droplet mode) implemented in the MOE2022.02 suite (Figure 8).

The docking analysis was carried out considering the lowest energy conformation of the corresponding CDs as rigid receptors, whereas the Se-NSAID derivatives were considered as flexible ligands. The poses obtained were clustered according to their final energy, and the resulting complexes were analyzed in order to evaluate the host–guest interactions. The best poses obtained for each compound were further studied in a molecular dynamic (MD) assay in order to assess the stability of the theoretical complexes modeled.

In this regard, an initial configuration for the complex compound-CD was constructed for derivative **I.3e**. As shown in Figure 9a, the naphthalene ring was incorporated into the inner cavity through the major face of the corresponding CD (β- or γ-CDs).

The initial complexes for indomethacin-derived compounds **I.4a–e** revealed that the chlorobenzene moiety was inserted in the inner cavity of the CD, whereas the indole ring and the side chain remained on the outside, near the major face (as shown by the representative example of Figure 9b).

Regarding the derivatives **II.1–5**, two different starting complexes were built. On the one hand, one of the lateral aromatic rings was incorporated into the cavity through the larger face; whilst on the second one, the central diacyl diselenide moiety was placed in the inner cavity. Herein, the aromatic lateral rings are located outside the major and minor faces. Moreover, the fact that the presence of Se atoms both in the side chains of selenoesters of series I and in the diacyl diselenide moieties of series II could enhance possible intermolecular interactions should be taken into consideration, as this element has empty *d* orbitals which would facilitate such interactions [44] and contribute to the stability of the complexes.

In this context, some representative examples of the best poses obtained for complexes formed by each CD and certain Se-NSAID derivatives are displayed in Figure 10, Figure 11, Figure 12 and Figure 13.

The data obtained in the docking assay in the presence of explicit solvent (Appendix A) showed that complexes with γ-CD were, in principle, more favorable than those proposed for β-CD, as the scoring values were less negative for the latter.

To further characterize the interactions of the most active derivative towards cancer (compound **II.5**) [32] with β-CD, an analysis of the chemical shifts of the NMR spectra for the protons of the β-CD was carried out using different concentrations of the compound (Appendix A). The magnitude of these shifts should be proportional to the intensity of the host–guest interaction. Thus, it can be inferred that hydrogens H_3_ and H_5_ present a similar interaction with **II.5**, both of them being greater than H_6_.

The complexes obtained in the docking assay were analyzed individually, aiming to identify the most significant interactions between the host and ligand, with special interest in those where the Se atoms present in the compounds may be involved. Subsequently, an MD strategy was applied, and the complexes were reanalyzed to study the interactions that have been modified.

As a preliminary indicative criterion of the stability of the complexes, the corresponding energy of the complexes was obtained. Thus, once the pose was selected, the energy of the system was calculated using the corresponding algorithm implemented in the MOE2022.02 suite. The subsequent forcefield equation was applied, specifying the solvent model (Born). The interior dielectric and solvent dielectric constant values taken were 80. Some representative data can be found in the Appendix A.

The data obtained for β-CD complexes revealed that compounds **I.3e** and **II.5** showed the best results both in terms of score and theoretical complex stability, followed by **I.4b** and **II.2**. In 87% of the poses obtained for derivative **I.3e** (Figure 10a), it could be observed that the nonpolar naphthalene ring remained inside the cavity, while the side chains, with more polar groups, remained exposed to the solvent. Thus, the methoxy group would be located through the minor CD face, while the Se carrier chain faced the major entry, respectively. Moreover, a H-arene interaction (represented in green in Figure 10a) established between one of the glucoses (GLC 5) of the β-CD and the naphthalene ring (C5) could be observed, while the N atom of the cyano group established numerous interactions with solvent molecules. A further MD assay revealed that the aforementioned H-arene interaction was maintained, whereas the N and Se atoms interacted with other GLC residues (GLC 1 and GLC 2) in a H-acceptor type interaction, as shown in Figure 10b.

Considering the study of compound **II.5** as a representative example of series II, 65% of the poses obtained (Figure 10c) showed that the diacyl diselenide moiety remained within the cavity, establishing two strong and stable interactions (H-donor type) between the Se atoms and two of the GLC residues (GLC 2 and GLC 3) of the β-CD. The rest of the molecule would be thus expanded on both the major and minor faces and would form H-arene type interactions between the lateral rings and the solvent molecules. As for the rest of the poses, the presence of one of the lateral rings was detected inside the inner cavity linked by a weak H-arene interaction, whilst the rest of the molecule remained on the outside, exposed to the polar solvent (Figure 10d). 

In the case of the indomethacin-derived compounds **I.4a–e**, effective interactions could only be observed for **I.4b**. Indeed, 91% of the poses obtained (Figure 11a) showed that the chlorobenzene ring was kept in the inner cavity by a strong H-arene type interaction established with GLC 7, while the rest of the molecule was oriented towards the outside, crossing the major CD face. Furthermore, an interaction between the Se atom of the side chain and the GLC 1 (H-donor type) residue of the β-CD was detected, along with another H-arene interaction between this same GLC 1 and the indole ring. Nonetheless, these two interactions disappeared after the MD test.

In the case of the indomethacin-derived molecule of series II, it could be observed that in 92% of the poses (Figure 11b), the chlorobenzene ring was kept within the cavity, being maintained by a strong H-arene type interaction between this chlorobenzene ring and the GLC 3 residue. Moreover, another interaction of this type was established between the indole ring and GLC 5. The rest of the structure was oriented across the major face and exposed to the solvent. These interactions further weakened according to the results obtained after the MD analysis. For the remaining 8% of the poses (Figure 11c), the diacyl diselenide group remained within the cavity, in which each Se atom interacted (H-donor type) with a GLC residue (GLC 3 and GLC 4, respectively). Thus, the two lateral indole rings and their corresponding decorations appeared to be located on both sides of the CD structure, highly exposed to the solvent. It should be noted that in general, the Se-NSAID derivatives that included an indole ring in their scaffold would locate this moiety outside the CD. This could be related to the existence of a certain steric hindrance that hindered the entry of the heterocycle into the inner cavity.

Although it was expected that the γ-CDs would form more effective guest–host inclusion complexes with the Se-NSAID derivatives due to the larger size of their inner cavity, only derivatives **I.3e** and **II.5** were able to form significant and stable interactions (Figure 12). Indeed, it was found in 90% of the analyzed poses that **I.3e** established a strong and stable interaction (H-donor type) between the Se atom and GLC 707 residue (Figure 12a). This Se-GLC interaction was also observed for **II.5** in 65% of the cases (Figure 12b), while in the other poses, this interaction was even reinforced by another one (H-acceptor type) established between the carbonyl group attached to the other Se atom and a second GLC residue (GLC 701), as shown in Figure 12c.

On the other hand, it should be noted that for derivatives **I.4a–e**, the indole ring remained in the inner cavity but failed to establish effective interactions with the GLUC environment, as can be seen in the representative example illustrated in Figure 13a.

Interestingly, the indole ring and the chlorobenzene that are attached partially to compound **II.2** remained within the cavity for 64% of the analyzed poses (Figure 13b), detecting interactions (H-donor type) established between GLC 701 and the carbonyl group linked to the benzene ring and the indole ring. These interactions were then weakened in the MD assay. For the remaining poses, it was observed that one of the Se atoms and the carbonyl group attached to another Se atom interacted with GLC 700 and GLC 703 residues (H-donor and H-acceptor types) (Figure 13c). These results were consistent with the data obtained for the above-mentioned electrostatic and interaction potential maps.

Although the analysis conducted so far was predominantly qualitative, it could be suggested that given the different conformational behaviors detected for the analyzed compounds depending on the solvation conditions of the assay (i.e., implicit or explicit solvent), this could influence the theoretical stability of the complexes. This influence was particularly pronounced for derivatives of series II.

By predominantly adopting the extended conformation over the folded one, a larger interaction surface is therefore exposed, both with GLC residues and solvent molecules, thus reinforcing or maintaining interactions, as can be seen in the case of compounds **I.3e** and **II.5**. Further studies involving quantitative MD would be necessary for obtaining and comparing the energies of the complexes before and after MD to confirm these findings.

Nevertheless, our preliminary results obtained both experimentally and by molecular modeling support the use of CDs for the improvement of solubility. These results are in accordance with recent published work in which β- and γ-CDs were used to encapsulate certain NSAIDs [45,46,47] and entail a framework for further developing novel formulations of Se-containing compounds with this type of carrier.

## 3. Materials and Methods

### 3.1. General Information

All the Se-NSAID derivatives evaluated in this study were obtained according to synthetic routes previously reported for the selenoesters of series I [31] and the diacyl diselenides of series II [32]. Likewise, the compounds were synthesized with a high grade of purity as they had been evaluated for their appealing anticancer potential in biological assays. Native β-CDs (≥97%) and γ-CDs (≥98%), with water contents of 12.51% and 7.52%, respectively, as determined using thermogravimetric analysis (TGA), were purchased from Sigma-Aldrich (Saint Louis, MO, USA).

The calculations for molecular modeling were performed on a SGI Virtu VS100 workstation equipped with MOE2022.02 and Discovery Studio v2.5 software packages.

### 3.2. Water Solubility Studies by ^1^H-NMR

The experimental formation of inclusion complexes with either β-CD or γ-CD was evaluated via quantitative ^1^H-NMR spectroscopy. This method is based on the consideration that the integration for a given signal that is selected as a marker for a given compound is proportional to its concentration [48]. Consequently, solutions of 0.5 mL in D_2_O (≥99.85% in deuterated component) with an excess of each compound were prepared, along with either 4 mg of dimethyl sulfone (as internal standard) or with the corresponding CD at 10 mM. ^1^H-NMR spectra were recorded at 298 K on a Bruker Avance Neo at a proton resonance frequency of 400 MHz. The solubilities were calculated considering the ratio between the area of the peaks belonging to the internal standard or the CDs and the different hydrogens of the Se-NSAID derivatives.

### 3.3. β- and γ-CDs Preparation and Analysis

The reference models AJUVEG [36] and CIWMIE10 [37] were obtained from the CSD database and were used as templates for building the initial models for the analyzed CDs. These models were imported into the MOE2022.02 workspace under a Generalized Born implicit solvent model (dielectric constant interior and exterior = 80). An implemented AMBER10_EHT force field and a mild minimization protocol (steepest descent rms = 0.1 kcal/mol/Å^2^ as completion criterion) were then applied.

The electrostatic map was calculated for the selected conformation by detecting the preferred regions for hydrogen bond acceptors or positive electrostatic potential at a potential value = −2 kcal/mol, and the regions for hydrogen bond donor or negative electrostatic potential at a potential value = −2 kcal/mol. The interaction potential map was calculated for the selected representative conformation by detecting the preferred regions for interaction with an OH2 probe at −5.5 kcal/mol and a dry probe at −2.5 kcal/mol.

### 3.4. Ligand Preparation and Analysis

The reference CSD models of the parent NSAIDs were exported to the MOE2020.02 workspace and modified to obtain the structure of the studied selenocompounds by using the implemented Builder tool. The templates used were ACSALA29 [39] for **II.1**, COYRUD [40] for naproxen-derived compounds **I.3e** and **II.3**, INDMET [41] for indomethacin-containing derivatives **I.4a–e** and **II.2**, KEMRUP [42] for **II.4**, and JEKNOC10 [43] for **II.5**.

The structures were first carefully curated in a Born implicit solvent model (dielectric = 80) under the AMBER10:EHT force field. A preliminary optimization was carried out with a root mean square gradient (RMS) of 0.1 kcal/mol/Å^2^ as the completion criterion. Restraints and constraints were not applied. These initial minimized conformations obtained for each compound were included in a database (SET1) and were considered as the starting point for the conformational analysis. Such analysis was carried out by means of the Conformational Search tool, implemented in the MOE2022.02 suite through a LowModeMD, with a rms = 0.001 and an energy window of 5 kcal. This approach generated conformations using a short ~1 ps run of MD at a constant temperature (300 K), followed by an all-atom energy minimization. Once the conformational trajectory was obtained, a representative lowest energy conformation for each ligand was selected and included in another database (SET2).

The study of conformational behavior was repeated using explicit solvent conditions (water, solvate routine implemented in the MOE2020.02 suite, spherical droplet mode, and explicit water molecules taking 2–4 Å as margin values). Once the conformational trajectory was obtained, a representative lowest energy conformation for each ligand was selected and incorporated into another database (SET3).

The logP, van der Waals volume and surface data for SET2 structures were calculated with the MOE2022.02 Calculate descriptors tool.

### 3.5. Docking

The corresponding CD models for β- or γ-CDs were selected as the receptors (host). The preparation protocol used could be summarized as follows: (a) import of the CD models to the MOE’s workspace; (b) examination of any errors such as clashes, missing atoms or bonds, and incorrect stereochemistry; (c) assignation of protonation states by means of the MOE’s Protonate3D protocol (pH = 7); (d) establishing the solvation conditions: explicit water conditions (solvate routine implemented in the MOE2020.02 suite, spherical droplet mode, and explicit water molecules taking 2–4 Å as margin values); and (e) minimization of the energy (gradient = 0.1 kcal/mol/Å^2^) to ensure that it is in a stable conformation by applying the AMBER10:EHT force field routine.

It should be noted that the structures selected as theoretical ligands integrated in the SET3 database were further minimized with the implemented MOPAC engine (PM3 semi-empirical approach).

The docking study was carried out by applying the Dock algorithm implemented in the MOE2022.02 suite (AMBER10:EHT force field). The procedure could be summed up as follows: (a) selection of the host (receptor) atoms (SITE) in the analyzed CD model; (b) selection of the ligand in the lowest energy conformation (SET3); (c) selection of the placement method: Triangle Matcher; and score method: London dG scoring, which estimates the free energy binding of the ligand from a given pose with 100 as the maximum pose number; (d) selection of the refinement method: Rigid Receptor, Score GBVI/WSA dG scoring, that estimates the free energy of binding of the ligand from a given pose with 5 as maximum pose number; (e) launching of the ligand database (SET3) to the active site; and (f) selection of the best poses and further clustering analysis of the results.

In addition, five independent docking processes were carried out. The final data evaluated were the mean of the final scoring value corresponding to the best pose obtained for each ligand in the five dockings carried out, i.e., the mean of five data for each ligand.

### 3.6. Molecular Dynamics

The stability of the final selected complexes was analyzed by means of an MD simulation using the Molecular Dynamics tool implemented in the MOE2022.02 suite. The NPA (Noisé–Poincaré–Andersen equations of motion) algorithm was selected. The simulation protocol consisted of a minimization segment (t = 10 ps) at 0 K; followed by a heating step (t = 100 ps) from 10 to 300 K; then an equilibration step (t = 100 ps) at 300 K; followed by a production step (t = 500 ps) at 300 K; and a final cooling step (t = 100 ps) from 300 to 10 K. Light bond (bonds involving H or Lone Pair, LP) were constrained. The CD backbone was limited during this process, using a fixed atom constraint. No tethered atoms were established. The time-step was 0.001, and the atom positions of the model were written to file every 500 steps.

## 4. Conclusions

Both the theoretical models used for the β- and γ-CDs and the theoretical conformational analysis used for the selenocompounds seem to be robust. Additionally, docking studies predicted the most stable host–guest interaction of both CDs with compounds **I.3e** and **II.5**. This theoretical outcome is supported by the experimental results obtained. In fact, the data depicted in this article revealed an ameliorated water solubility for these complexes compared to compounds **I.3e** and **II.5** of almost two orders of magnitude. This increase could be ascribed to the fact that these two compounds might form the most stable interactions with the studied CDs, as predicted by theoretical calculations. Despite these promising results, it should be noted that this is a preliminary study; it may suggest the use of CDs-based complexes to improve the solubility and bioavailability of Se-NSAID derivatives, but nevertheless, more studies would be needed to test the full potential of these carriers along with the selenocompounds evaluated in this work.

## Figures and Tables

**Figure 1 ijms-25-01532-f001:**
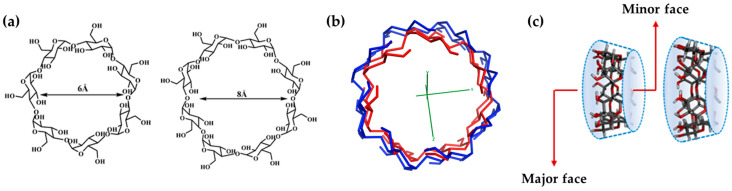
Differences in size among the cyclodextrins used in this study: (**a**) scheme of β-CD (**left**) and γ-CD (**right**) showing the inner diameter; (**b**) overlay of β-CD (red) and γ-CD (blue); (**c**) scheme of β-CD (**left**) and γ-CD (**right**) toroids.

**Figure 2 ijms-25-01532-f002:**
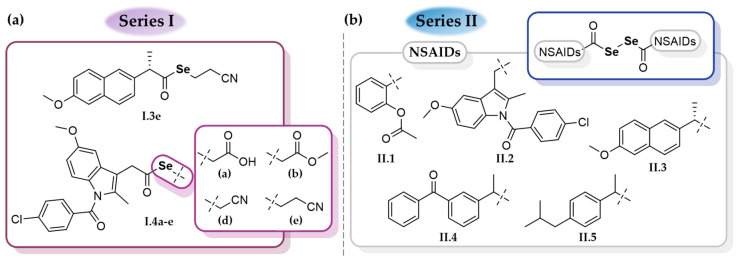
General structures of the Se-NSAID derivatives based on (**a**) selenoesters (series I) and (**b**) diacyl diselenides (series II) included in this study.

**Figure 3 ijms-25-01532-f003:**
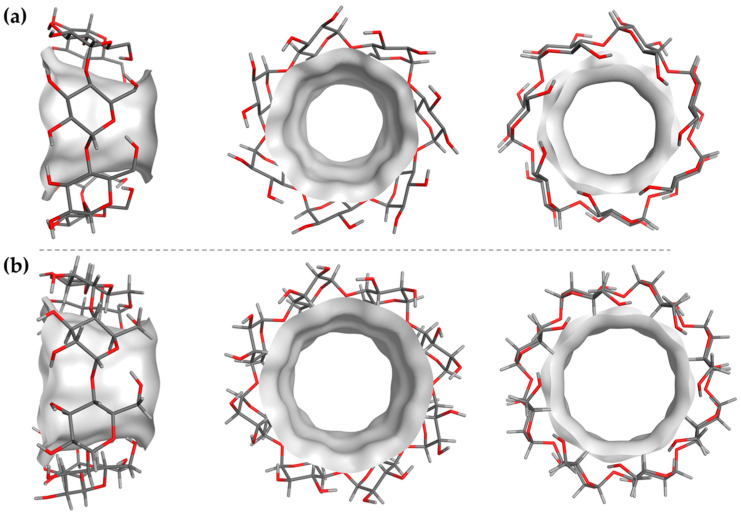
Binding site (inner cavity) for (**a**) β-CD and (**b**) γ-CD (inner surface in solid light grey; CD structure in sticks; C in grey; H in light grey; O in red).

**Figure 4 ijms-25-01532-f004:**
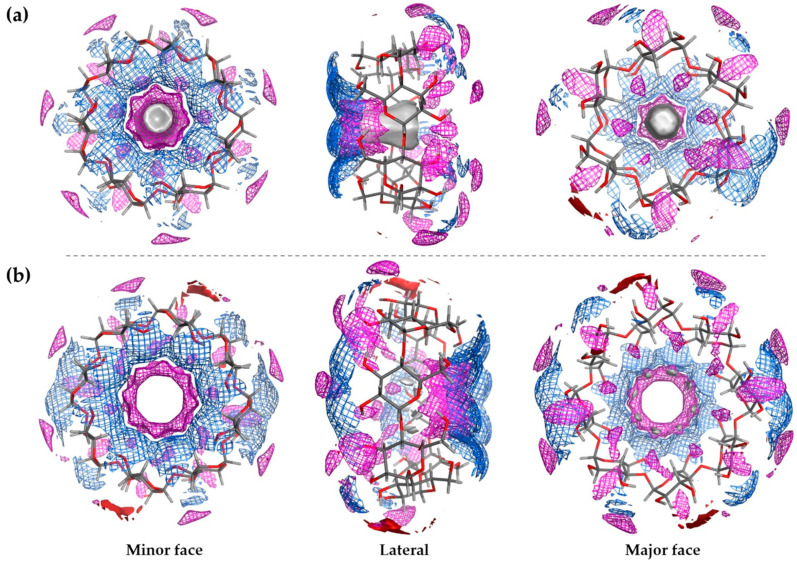
Electrostatic and interaction potential maps for (**a**) β-CD and (**b**) γ-CD. Red solid: regions for hydrogen bond acceptors or positive electrostatic potential at a potential value = −2 kcal/mol. Blue line: regions for hydrogen bond donor or negative electrostatic potential at a potential value = −2 kcal/mol. Grey solid: hydrophobic region at a potential value = −2.4 kcal/mol. Magenta line: interaction potential with an OH2 probe at −5.5 kcal/mol. CD structure in sticks (C in grey; H in light grey; O in red).

**Figure 5 ijms-25-01532-f005:**
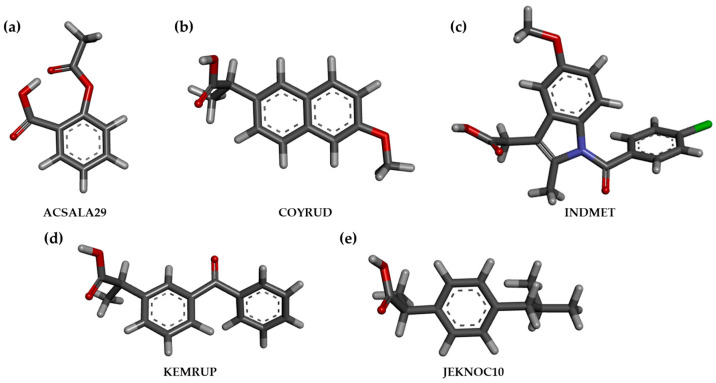
Reference crystallographic models taken as templates for the construction of the analyzed derivatives, showing their CSD reference for (**a**) aspirin [39], (**b**) naproxen [40], (**c**) indomethacin [41], (**d**) ketoprofen [42], and (**e**) ibuprofen [43], respectively (C in dark grey; H in light grey; O in red; N in blue; CI in green).

**Figure 6 ijms-25-01532-f006:**
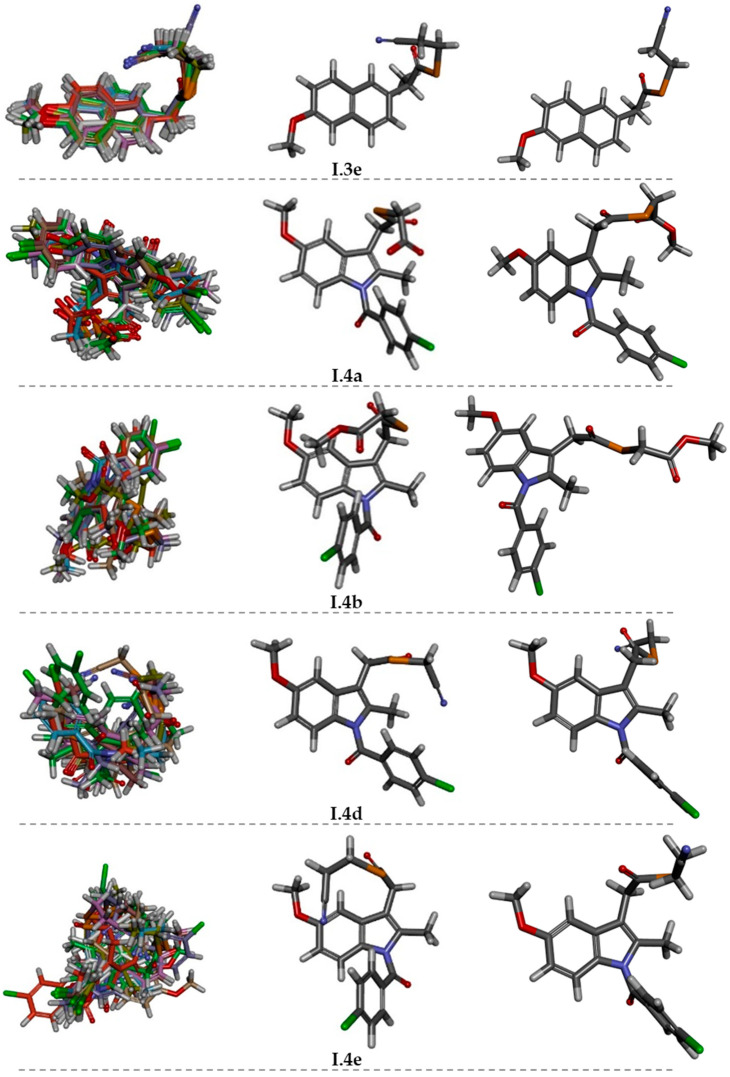
Left: some representative examples of conformational behavior for **I.3e** and **I.4a–e** compounds (implicit solvent, molecular overlay taking 100% steric as alignment criteria). Center: lowest energy conformation (implicit solvent). Right: lowest energy conformation (explicit solvent). Magenta dotted line: π–π intramolecular interaction (C in dark grey; H in light grey; O in red; N in blue; Se in orange; CI in green).

**Figure 7 ijms-25-01532-f007:**
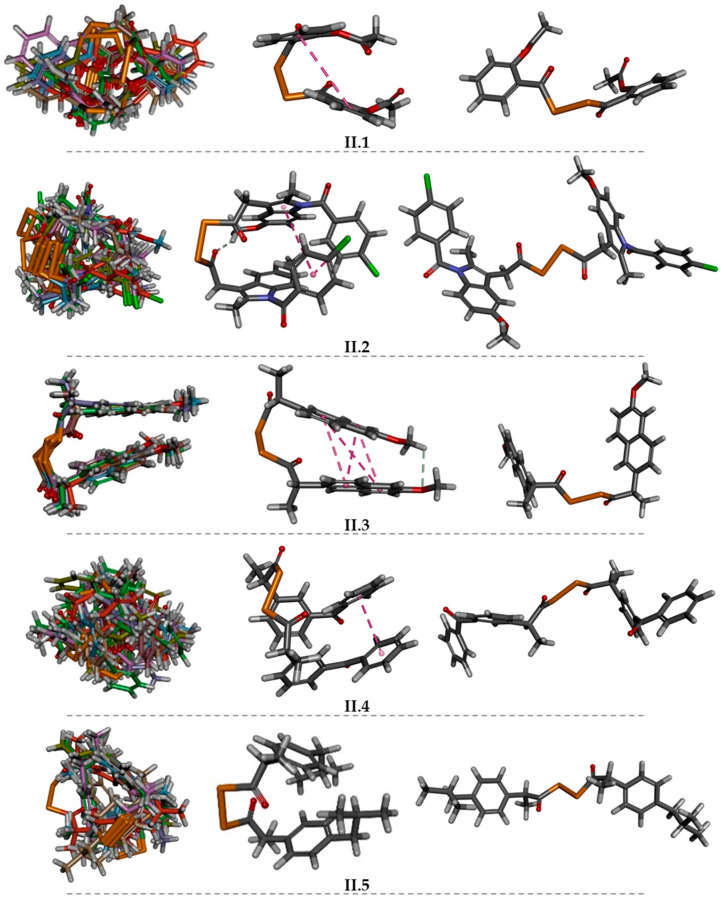
Left: some representative examples of conformational behavior for **II.1–5** compounds (implicit solvent, molecular overlay taking 100% steric as alignment criteria). Center: lowest energy conformation (implicit solvent). Right: lowest energy conformation (explicit solvent). Magenta dotted line: π–π intramolecular interaction (C in dark grey; H in light grey; O in red; N in blue; Se in orange; CI in green).

**Figure 8 ijms-25-01532-f008:**
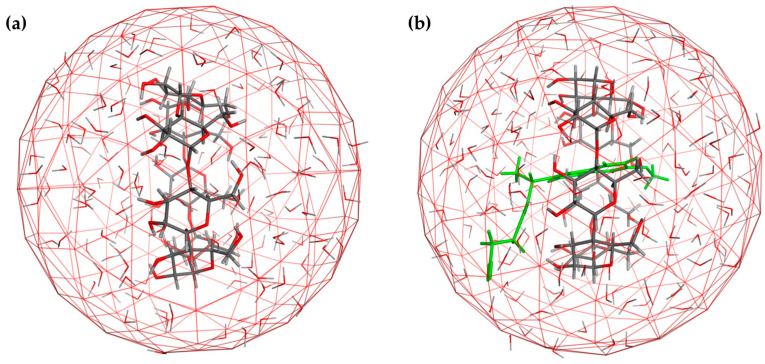
Representative docking strategy example under explicit solvent (water) droplet: (**a**) β-CD and (**b**) inclusion complex of β-CD and compound **I.3e**. (**a**) C in dark grey; H in light grey; O in red; N in blue; Se in orange; CI in green (**b**) compound **I.3e** in green.

**Figure 9 ijms-25-01532-f009:**
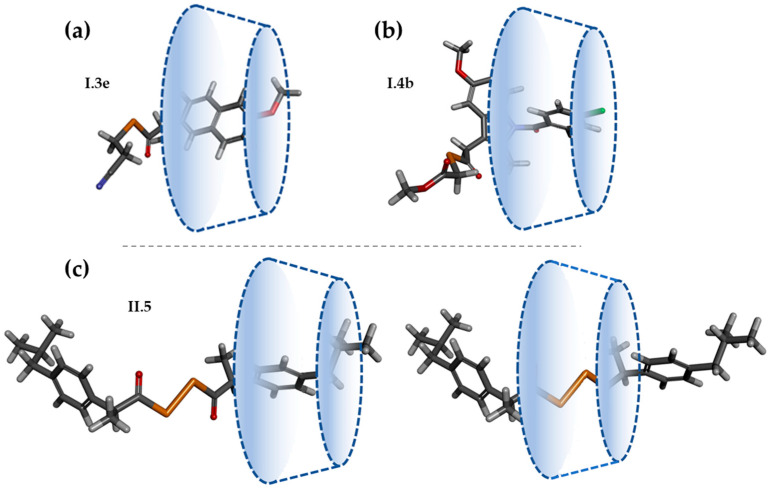
Some examples of initial complexes built for the docking with (**a**) the central naphthalene ring (compound **I.3e**), (**b**) the chlorobenzene ring (compound **I.4b**), and (**c**) the isobutylbenzene ring or diselenide moiety (**II.5**) placed inside the CD cavity (C in dark grey; H in light grey; O in red; N in blue; Se in orange; CI in green).

**Figure 10 ijms-25-01532-f010:**
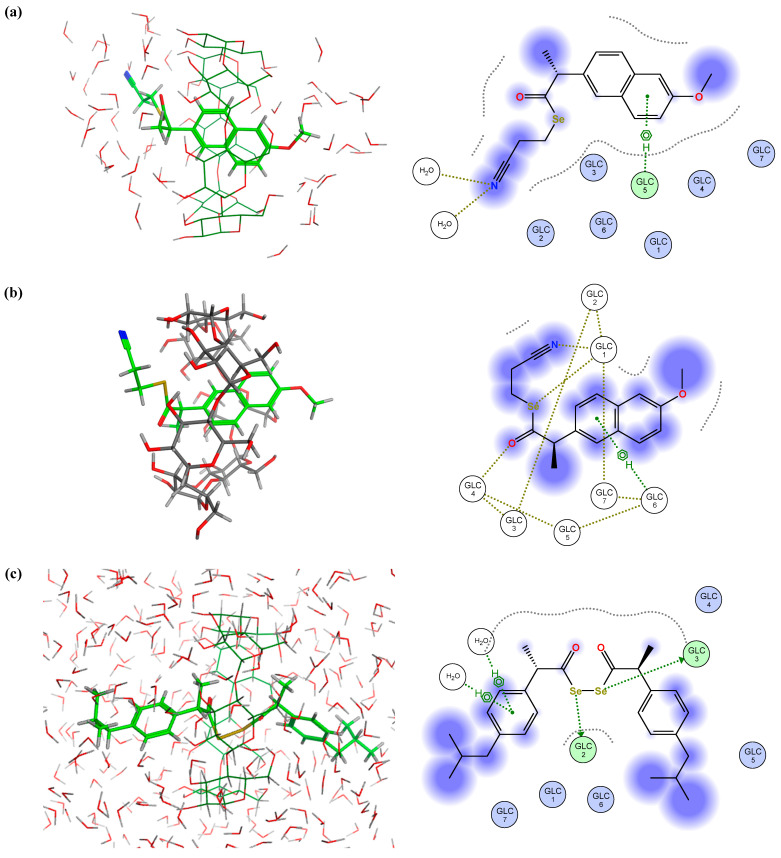
Representation of the best poses for β-CD and (**a**) compound **I.3e**; (**b**) **I.3e** after MD assay (solvent molecules are hidden for easy visualization); (**c**) and (**d**) compound **II.5** in two different positions, respectively.

**Figure 11 ijms-25-01532-f011:**
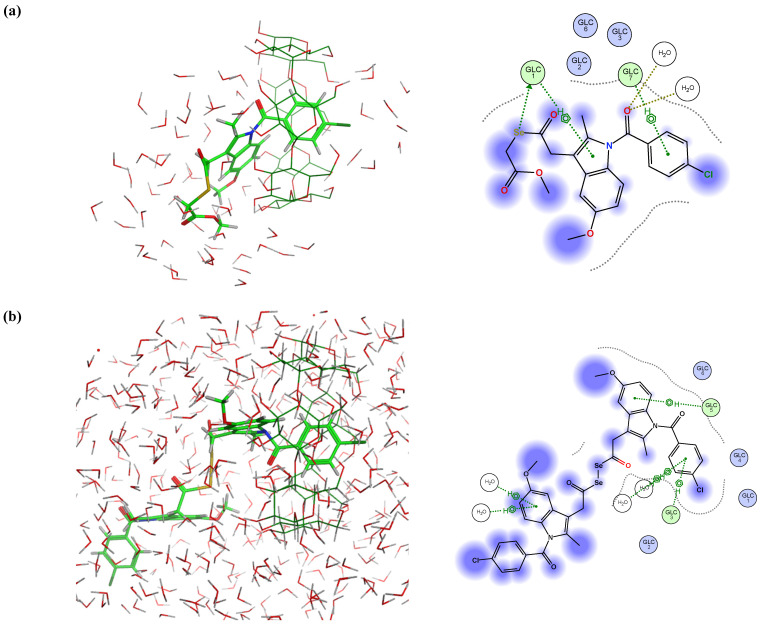
Representative best poses for complexes of β-CD and (**a**) compound **I.4b**; (**b**) and (**c**) compound **II.2** in two different positions, respectively. (Host: C in dark green; H in light grey; O in red. Solvent: H in light grey; O in red. Ligand: C in green, H in light grey; O in red; N in blue; Se in orange; Cl in dark green).

**Figure 12 ijms-25-01532-f012:**
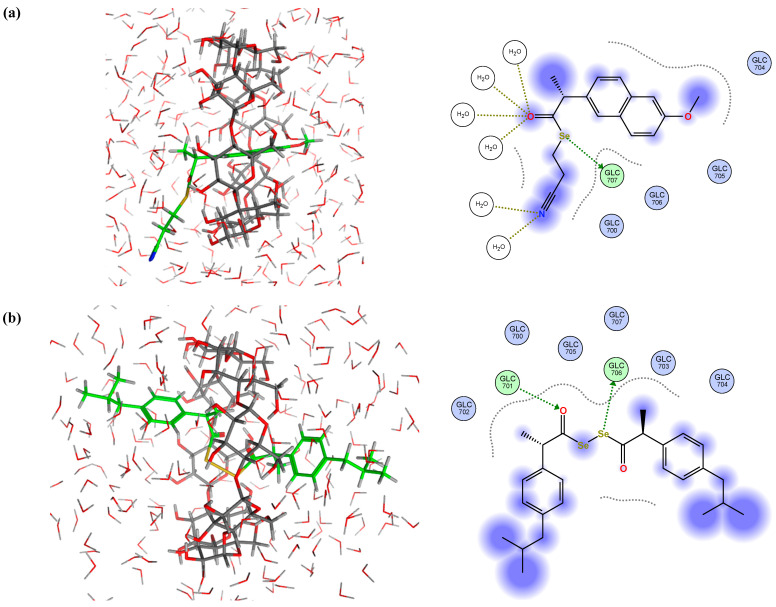
Representative best poses for complexes of γ-CD and (**a**) compound **I.3e**; (**b**) and (**c**) compound **II.5** in two different positions, respectively. (Host: C in dark green; H in light grey; O in red. Solvent: H in light grey; O in red. Ligand: C in green, H in light grey; O in red; N in blue; Se in orange; Cl in dark green).

**Figure 13 ijms-25-01532-f013:**
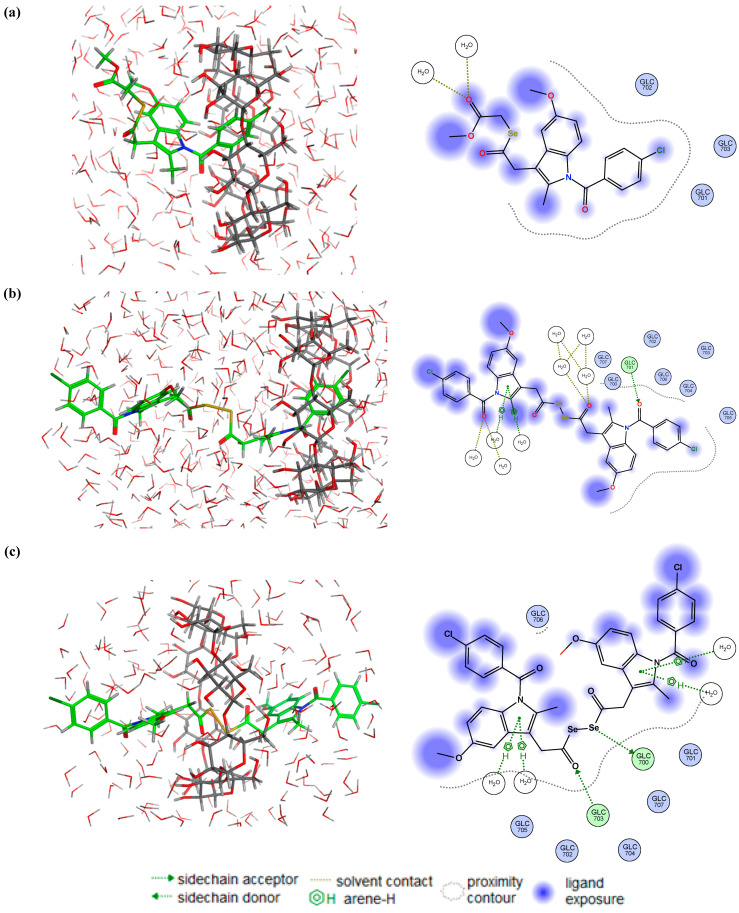
Representative best poses for complexes of γ-CD and (**a**) compound **I.4b**; (**b**,**c**) compound **II.2** in two different positions, respectively.

**Table 1 ijms-25-01532-t001:** Solubility of Se-NSAID derivatives with either β- or γ-CDs in D_2_O.

Ref.	Water Solubility (M)		
	Compound	Compound + β-CD	Compound + γ-CD
**I.3e**	9.70 × 10^−6^	9.92 × 10^−4^	2.73 × 10^−4^
**I.4a**	2.24 × 10^−5^	5.65 × 10^−5^	5.24 × 10^−5^
**I.4b**	1.20 × 10^−5^	1.57 × 10^−4^	4.39 × 10^−5^
**I.4d**	2.00 × 10^−5^	4.53 × 10^−5^	3.37 × 10^−5^
**I.4e**	5.72 × 10^−6^	4.55 × 10^−5^	4.21 × 10^−5^
**II.1**	8.43 × 10^−6^	1.01 × 10^−4^	1.51 × 10^−5^
**II.2**	1.32 × 10^−7^	7.96 × 10^−6^	7.60 × 10^−6^
**II.3**	6.05 × 10^−7^	2.19 × 10^−7^	6.14 × 10^−6^
**II.4**	5.57 × 10^−6^	2.70 × 10^−5^	1.77 × 10^−5^
**II.5**	2.12 × 10^−5^	5.42 × 10^−3^	7.50 × 10^−3^

**Table 2 ijms-25-01532-t002:** Descriptors of the Se-NSAID derivatives analyzed.

Ref.	LogP	Vol ^a^ (Å^3^)	VSA ^b^ (Å^2^)
**I.3e**	3.56	300.75	336.28
**I.4a**	4.77	382.25	417.70
**I.4b**	5.44	402.38	435.27
**I.4d**	5.47	381.50	416.68
**I.4e**	5.50	395.50	430.41
**II.1**	5.02	341.38	360.93
**II.2**	10.87	671.50	725.31
**II.3**	6.99	476.63	510.56
**II.4**	8.40	526.13	557.23
**II.5**	9.04	474.13	523.38

^a^ van der Waals volume. ^b^ van der Waals surface.

## Data Availability

The data presented in this study are available on reasonable request from the corresponding author.

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
