# Peer review of "Formulation Studies with Cyclodextrins for Novel Selenium NSAID Derivatives"

_ijms, 2024, doi:10.3390/ijms25031532_

Round 1

Reviewer 1 Report

Comments and Suggestions for Authors

The manuscript titled "Formulation studies with cyclodextrins for novel selenium NSAID derivatives" by Ramos‐Inza et al. presents an in-depth investigation into the formulation of inclusion complexes (ICs) with cyclodextrins (CDs) to enhance the water solubility, stability, and bioavailability of selenium-containing nonsteroidal anti-inflammatory drug (NSAID) derivatives with demonstrated anticancer activity.

Major Revisions:

1. The abstract provides a clear overview of the study but lacks specific information about the results and implications. Include key findings and their significance in the abstract.

2. Clarify the objectives of the study in the introduction section.

Provide more background on the challenges associated with selenium-containing compounds and the potential benefits of cyclodextrins in overcoming these challenges.

3. Introduce a section comparing the findings of this study with recent research in the field [e.g. 10.1016/j.molliq.2022.119548 and 10.3390/pharmaceutics15010071 etc]. Highlight how this study contributes to or aligns with existing knowledge. Reference and briefly discuss relevant recent studies.

4. Emphasize the key factors influencing the choice of β and γ-CD, supported by molecular modeling and experimental results.

Elaborate on the impact of the conformational behavior of Se-NSAID derivatives on their interaction with CDs.

5.Offer a more comprehensive explanation of the docking results, highlighting specific interactions between Se-NSAID derivatives and cyclodextrins.

Discuss the implications of the conformational changes observed during the molecular dynamics assay on the stability of the complexes.

Addressing these major revisions will enhance the manuscript's clarity, completeness, and overall impact, providing readers with a more insightful understanding of the study's findings and their potential applications.

Reviewer 2 Report

Comments and Suggestions for Authors

The manuscript lacks 'Supplementary Information', which shows the NMR spectra and also gives the xyz coordinates of the complexes discussed. The introduction mentions that the article is about the structural characterisation of CD complexes "we study the structural chemistry of CD complexes", while the manuscript only discusses this topic theoretically. Furthermore, no theoretical details, such as the energies of the complexes, are provided and there is no discussion of the relationship between stability and the reported solubility results of the complexes. Any evaluation of the manuscript due to the lack of data is impossible.

Author Response

"Please see the attachment

Round 2

Reviewer 1 Report

Comments and Suggestions for Authors

The manuscript titled "Formulation Studies with Cyclodextrins for Novel Selenium NSAID Derivatives" by Sandra Ramos-Inza et al. provides a thorough investigation into the use of cyclodextrins (CDs) to address pharmaceutical limitations associated with selenium-containing compounds, particularly focusing on their potential as antitumoral agents.

The study delves into the structural chemistry of CD complexes formed with a diverse set of nonsteroidal anti-inflammatory drugs (NSAID) combined with selenium in various chemical forms. The authors use nuclear magnetic resonance (NMR) spectroscopic techniques and molecular modeling to analyze these complexes, shedding light on the molecular interactions and aiding in the selection of suitable CDs for the intended applications.

In conclusion, this manuscript significantly contributes to the field of cyclodextrin-based formulations, particularly in the context of selenium-containing compounds with antitumoral activity. The combination of experimental techniques and molecular modeling, along with the comprehensive analysis of CD complexes, makes this work a valuable resource for researchers in molecular sciences.

Reviewer 2 Report

Comments and Suggestions for Authors

The manuscript requires additional revision (the previous recommendations have only been met fragmentarily). I strongly recommend that the following points be discussed and reconsidered:

1.       Please provide the x,y,z-coordinates of the most stable complexes shown and discussed in the manuscript. The majority of the manuscript focuses on computation results. Therefore, I strongly recommend including detailed information on the structures of the complexes by providing their x,y,z-coordinates in  the 'Supplementary Information'.

2.       Please provide the energies of the complexes – the authors discuss the theoretical stability of the complexes (page 12, lines: 304-305), without providing values for their energies. Please explain how the theoretical stability of the complexes was assessed.  

 I understand that the theoretical calculations are preliminary, but if the theoretical stability of complexes is discussed, it should be clear to the reader what data is being referred to.

3.       Please provide a comprehensive analysis of NMR spectra as evidence for a) the formation of inclusion complexes (based on chemical shift analysis) and b) the stoichiometry of 1:1 complexes (as only 1:1 complexes were considered in the computation studies).  

4.       Please correct the reference 40 on page 2, line 54.

5.       Please provide a description of the logP descriptor.

6.       Please discuss how the different solubilities of cyclodextrins (b and g) in water may affect the solubility of their complexes?  

While the manuscript exhibits great potential, it necessitates extensive revision.
